# Sustainability Considerations of Green Buildings: A Detailed Overview on Current Advancements and Future Considerations

**Tianqi Liu** [1,†], **Lin Chen** [2,†], **Mingyu Yang** [3,†], **Malindu Sandanayake** [4,*], **Pengyun Miao** [1], **Yang Shi** [5] **and Pow-Seng Yap** [1,*]

1   Department of Civil Engineering, Xi'an Jiaotong-Liverpool University, Suzhou 215123, China
2   School of Civil Engineering, Chongqing University, Chongqing 400045, China
3   School of Materials Science Engineering, Nanjing University of Science and Technology, Nanjing 210094, China
4   College of Engineering & Science, Victoria University, Melbourne, VIC 3011, Australia
5   Department of Architecture and Design, Xi'an Jiaotong-Liverpool University, Suzhou 215123, China
*   Correspondence: malindu.sandanayake@vu.edu.au (M.S.); powseng.yap@xjtlu.edu.cn (P.-S.Y.)
†   These authors contributed equally to the work.

**Abstract:** The concept of green building has gradually formed with the increase in public awareness of environmental protection, which also covers a wide range of elements. The green building is the fundamental platform of sustainable development. This review paper provides solutions for the multi-dimensional and balanced development of green building. Since green building is the development trend of the construction industry, it presents an opportunity to mitigate global warming and accomplish energy efficiency. However, the problem is that the development of green building's implementation is restricted by the lack of government policies, imperfect technical abilities and unreasonable economic benefits. One conclusion drawn from the results shows that the benefits of green building implementation include environmental, economic, social, and health and safety aspects. Moreover, it is crucial to improve the awareness of stakeholders to promote the development process of green building. The government should launch campaigns to encourage developers and tenants to embrace green building, which can add value to buildings. The novelty of the paper provides a more systematic review on the sustainable considerations of green building than previous efforts in the literature. Bibliometric analysis is conducted through VOS viewer software. This review paperdiscusses the relevant benefits and challenges of green building through a critical review of existing research knowledge related to green building. The current advancements in green building are highlighted in this paper. Importantly, future recommendations for standards and policy formulation and future research directions are proposed in this review article.

**Keywords:** green building; sustainable development goals; environmental impacts; economic impacts; social impacts

## 1. Introduction

Cities accommodate more than 50% of the world's population and facilitate a large proportion of the important economic activities of a country [1]. The high concentration of economic activities and population distribution in urban regions has an influence on the environment [1]. Massive increases in carbon dioxide emissions due to human activities have further widened the gap between actual emissions and the goal of controlling global warming. Global carbon dioxide emissions have increased to 35.6 billion tonnes in 2012, which includes 28% from China, 16% from the United States, 11% from the European Union, and 7% from India [2]. Buildings are the largest man-made entities that contribute to huge amounts of carbon emissions. In addition, buildings are the largest energy-consuming asset in a city, accounting for 40% of total global energy consumption [1,3–6]. Therefore, reducing life cycle carbon emissions and the energy consumption of buildings is the

key to reducing the impact on the environment, economy and society and achieving sustainable development goals (SDGs). Building emissions mainly refer to the greenhouse gas produced as a result of resource consumption across the whole life cycle of a building, and can therefore relate to the daily life and work of the building's inhabitants [2]. Even a slight reduction in energy consumption and carbon emissions in buildings can influence the positive and sustainable lifestyles of its inhabitants. Therefore, it is a contemporary requirement to investigate every possibility of reducing the environmental, economic, and social impacts of buildings over their life cycle.

"Green building" is one such concept that has been introduced to reduce these environmental burdens of buildings over their life cycle. Green buildings have higher commercial value compared to traditional buildings, due to the perceived low carbon emissions, energy savings, and maximized economic benefits throughout the life cycle [7–11]. Therefore, green building could be a mandatory design feature in future city architecture designs to minimize environmental impacts and maximize economic and social benefits [1,12,13]. A green building is also categorized as a climate-resilient building wherever possible uses appropriate technologies to reduce energy consumption, while using locally available recyclable construction materials to achieve the lowest costs [2,14–17]. Moreover, green renovated buildings could also obtain important benefits in energy and carbon dioxide reduction aspects [1,18,19].

The objective of this review paper is to present a comprehensive review on considerations of green building that compliments sustainable development goals. The aim of this paper is to provide a focused review on the recent advances in green building, its research gap, the associated challenges and possible solutions to improve the current situation of this research area. The definition and research methodology are shown in this paper as well as findings and discussions. Especially, the research gap in current green building development will be discussed in Section 2. Current advancements in green building are also highlighted. Furthermore, this review paper can offer a comprehensive reference for overcoming the obstacles to the implementation of green building for future academic researchers.

## 2. Definition of a Green Building and Research Gap

A green building is often defined as an energy-saving building, ecological building or sustainable building [2,20,21]. However, there are many differences in green building definitions. American architects Paola Soleri and Ian Lennox McHarg stated that the concept of "green" is used to emphasize people-oriented and sustainable development to realize harmonious symbiosis among human, architecture and nature [2]. It has been further stated that ecological systems and the natural environment have become design considerations in the early stages of construction, which is the root of green building. Several researchers have attempted to define what a green building is in order to mitigate misinterpretation of green buildings [6,22–24]. Some scholars have identified a green building as a sustainable building or an intelligent building. According to [22], the importance of distinguishing the notions "Green", "Intelligent", and "Sustainable" was highlighted. The green building is more related to optimizing designs by adopting renewable energies, passive building design technologies, as well as scientific and systematic waste management techniques to minimize waste diversion to landfills [22]. The sustainable building can be regarded as an integral design, which more focuses on balancing environmental, economic, and social benefits over the life cycle of the asset [25]. Dwaikat and Ali [26] also claimed that sustainable building and green building can be used as interchangeable terms based on the scope, objective and context of the design, construction and operation of the building. However, there is still no unanimous definition of green buildings that is accepted by researchers and industry stakeholders across the world. Some definitions that have been proposed by organizations and individual researchers are provided in Table 1. Based on the content and key characteristics of the green building described, definitions can be classified into three categories, which are general definitions (No. 1 and No. 10), energy and resource-

focused definitions (No. 2 and No. 9), and comprehensive definitions (No. 3, No. 4, No. 5, No. 6, No. 7, No. 8). Through the summary, it can be seen that green building is not a simple concept that focuses on the building itself, but a comprehensive idea concerning people, society, the economy, and the environment through the entire life cycle of the building. Hence, designing a green building requires efforts from all involved sectors.

**Table 1.** Green building definitions and features.

| No | Country | Defining Body/Person | Definition/Key Features of a Green Building | References |
|----|---------|----------------------|---------------------------------------------|------------|
| 1 | | World Green Building Council (WGBC) | Reduce negative impact, enhance positive impact.<br>• Negative: design, construction, and operation stages;<br>• Positive: climate and natural environment. | [27] |
| 2 | | U.S. Environmental Protection Agency (EPA) | An environmental friendly structure creation practice.<br>• Be resource efficient throughout the entire life cycle. | [27] |
| 3 | USA | U.S. Green Building Council (USGBC) | Considering several key green building assessments during the entire life cycle of a building such as planning, design, and construction stages.<br>• Key assessments: (1) energy use, (2) water use, (3) indoor environmental quality, (4) material use, (5) and the building's effects. | [28] |
| 4 | | American Society of Heating, Refrigerating, and Air Conditioning Engineers (ASHRAE) | Considering the entire life cycle of buildings and high performance.<br>• Reduce natural resource consumption.<br>• Diminish emissions that have negative impact on the indoor environment and atmosphere.<br>• Minimize solid and liquid waste.<br>• Reduce negative impact on site ecosystem. | [29] |
| 5 | | Kibert | Ecologically based principles (resource efficient). | [30] |
| 6 | UK | Building Research Establishment | • Enhance people's quality of life<br>• Conserve natural resources<br>• Promote company profit. | [27] |
| 7 | China | Ministry of Housing and Urban-Rural Development of the People's Republic of China (MOHURD) | • Sustainability<br>• Energy efficiency<br>• Comfort of human<br>• Safety and durability<br>• Health and comfortability<br>• Life convenience<br>• Resource efficiency<br>• Environmentally livable | [27] |

**Table 1.** *Cont.*

| No | Country | Defining Body/Person | Definition/Key Features of a Green Building | References |
|---|---|---|---|---|
| 8 | Singapore | Inter-Ministerial Committee on Sustainable Development (IMCSD) | From the perspective of energy and resource consumption and human feeling.<br>• Achieve energy and water efficiency by using eco-material for construction<br>• Use green spaces to provide a high-quality and comfortable indoor environment. | [27] |
| 9 | Japan | Architectural Institute of Japan | • Save energy and resources<br>• Adopt reuse and recycle materials<br>• Reduce the emission of toxic substances.<br>• Be harmonized with local environment, climate, culture, and tradition.<br>• Improve human quality of life while avoiding sacrificing the ecosystem's capacity. | [31] |
| 10 | Australia | Green Building Council Australia (GBCA) | • Meet the current need while saving construction costs | [29] |

Although the number of green building research papers has been increasing over the decades, there is still a research gap in the study of green building. Many research papers only focus on one or two aspects of green building areas. A systematic review has not been conducted in these areas until now. This paper will fill this gap to show a comprehensive understanding of green building areas. The importance of this paper lies in its guiding significance for the realization of the sustainable development goal.

### 3. Research Methodology

In order to precisely review the current barriers, progress, and future research directions for sustainable development of green buildings, the research process of this review paper is divided into five main steps, as shown in Figure 1. Firstly, the WoS (Web of Science) was chosen as the primary database for this review paper, followed by inputting the top five keywords of the research topic and initially obtaining over 1000 pieces of literature. The abstract and full text of the literature was read according to the research topic of this review paper, and 150 articles were finally identified. Based on the selected literature, a visual analysis of the literature was conducted along with an analysis of the content of the literature, resulting in the current development of the green building field and recommendations to promote the sustainable development of green buildings.

This review paper looks at literature related to the field of green building with data obtained from the WoS core collection database, which contains over 10,000 disciplines and is a fairly authoritative and comprehensive database [32]. Although Scopus has a broader collection of data, its data have a relatively high duplication rate compared to WoS [33]. In addition, studies have found no significant differences in the analysis results when using the two databases were used together [33]. Therefore, this review paper has used WoS as a valid database to conduct the review study.

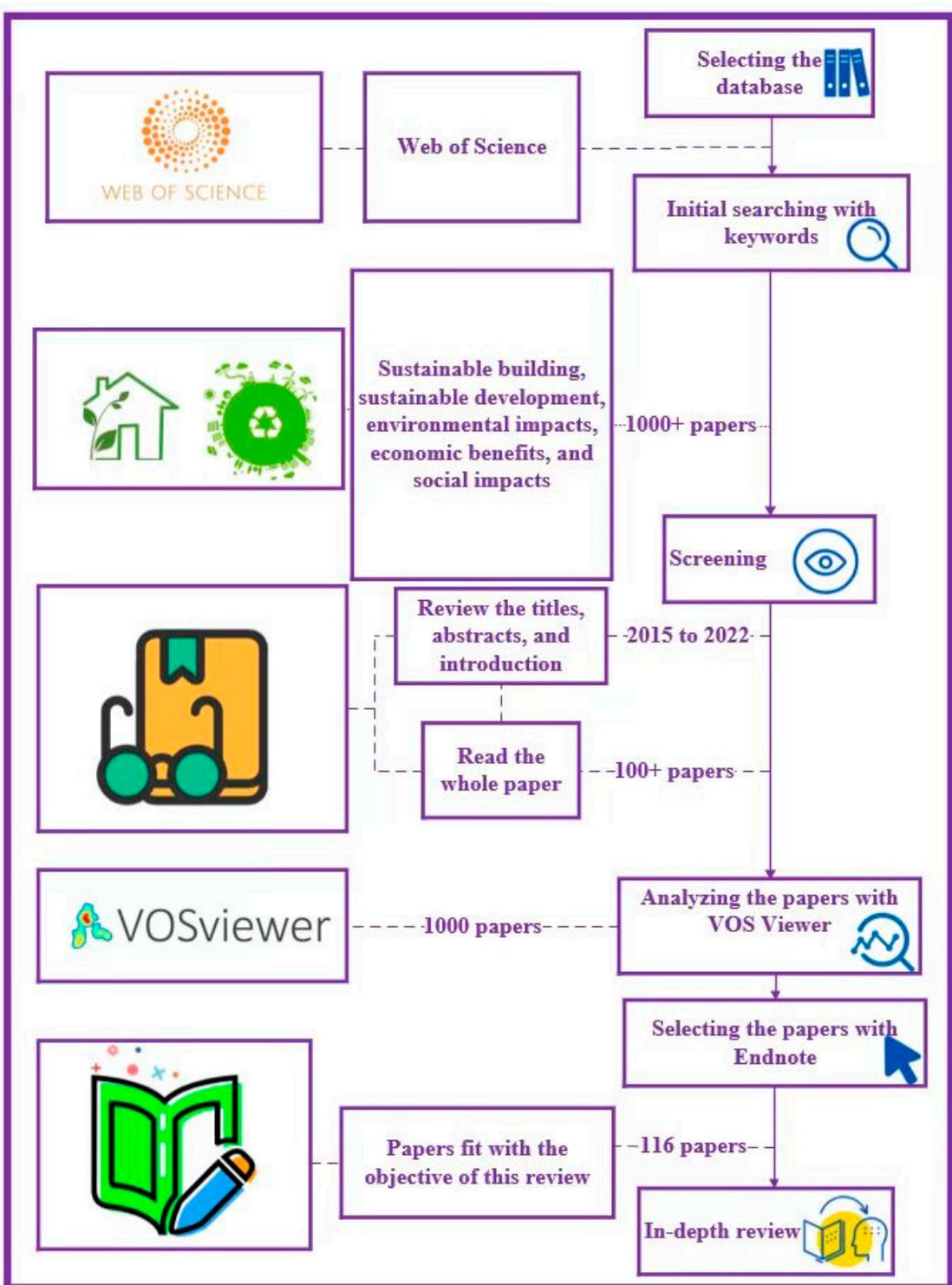

**Figure 1.** Research methodology process.

The field of green building has been extensively researched, the number of keywords used in the literature search was huge, and the number of studies retrieved as a result was also very large. According to the research theme of this review paper, five keywords were used; green building, sustainable building, environmental impacts, economic benefits, and social impacts were linked with the "AND" operator to capture the maximum results. Based on the keyword search, there were initially 1365 search papers; in order to find the most recent state-of-the-art research in the field of sustainable building, the time frame for the search was set to 2015–2022. Finally, by reading the abstracts, introductions, and

conclusions of the papers, those with obvious conclusions about sustainable building research, closely related to the five keywords, and highly relevant to the topic of this review were manually selected and started to be read carefully. VOS viewer is a free JAVA-based software developed in 2009 by Van Eck and Waltman from the Centre for Scientific and Technological Research at Leiden University in the Netherlands, intended to visualize literature data using analysis of unimodal undirected networks [34]. The current study used VOS viewer for visual analysis, which was effectively used to display the network structure of a research topic and effectively represent its information landscape, enabling the analysis and frontier detection of potential evolutionary mechanisms in a research area and suggesting research hotspots and trends in that area. In this review paper, the initial data were first derived from a keyword search; then, the data on the 1000 most-cited papers were exported and imported into VOS viewer to understand research hotspots. Then, the abstracts and conclusions of the paper were read using Endnote software, retaining those papers that fit the theme of this review paper and clicking on the hyperlinks to download them. Finally, the downloaded papers were read closely and used as references for this review paper.

### 4. Findings and Discussion

Figure 2 illustrates the hot topics related to the study of green and sustainable buildings. With sustainable development, it is important to consider the three elements of sustainability: environmental, economic, and social sustainability. Therefore, the environmental, economic, and social impacts of the implementation of green buildings should also be considered when investigating the development of green, sustainable buildings.

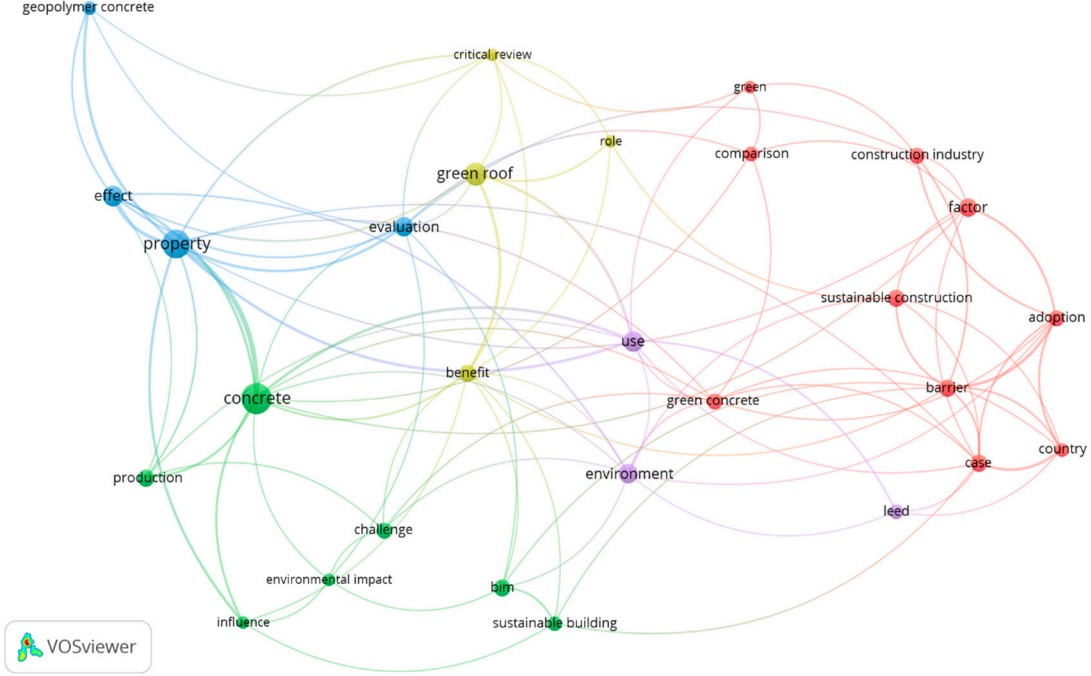

**Figure 2.** Research hotspots for green buildings by VOS viewer.

In the results of the VOS viewer network visualization, the hot terms are represented by their labels as well as circles, and the labels of the hot terms as well as the size of the circles are determined by the weight of the hot terms in the field of green building. The higher the weight of a hot term, the larger the label and the circle. The color of a buzzword is determined by the cluster to which the term belongs, meaning that buzzwords of the same color belong to the same cluster. The results of the literature measurement assessment indicate that the main buzzwords in the field of green building can be summarized as follows: the environmental impact of green building, the economic impact of green building,

the social impact of green building, and barriers and challenges to green building. Among them, the hotspots about environment in the VOS viewer network visualization results figure mainly include: environment, environmental impact, green roof, green concrete, sustainable construction, Leadership in Energy and Environmental Design (LEED), and green. The hotspots about economy in the VOS viewer network visualization results figure mainly include: manufacture, use, property, evaluation, factor, and adoption. The hotspots about society in the VOS viewer network visualization results figure mainly include: construction industry, role, case, country.

Based on these keywords, it is further mentioned that the impact of green building on the environment is two-sided and it is necessary to investigate the specific impact of green building implementation on the environment. In addition, the assessment criteria of whether green buildings are economically sustainable depend on whether green buildings can bring economic benefits to the whole industry, so it is also important to explore the impact of green buildings on economic development. At the same time, the capabilities of green building and manufacturing of green building both point to the social component of the sustainability triad, and are integral to the investigation of the social impacts of implementing green buildings. In the meanwhile, the bibliometric results suggest that the implementation of green and sustainable buildings can have beneficial effects on the human body, but it must be acknowledged that the implementation of green buildings can also present barriers and challenges in terms of institutions, policy and technology. As research has shown that many studies related to green building do not consider environmental, economic, social, health and safety aspects in an integrated manner, this review focuses on these buzzwords and explores the benefits and challenges of green building in a comprehensive and systematic manner based on the current state of research. Based on a critical thinking perspective, a comprehensive and systematic approach to sorting out the benefits and challenges of green building implementation could well clarify the specific impact of green building on sustainable development and provide valuable guidance for the future direction of green building development.

### 4.1. Benefits of Green Building Implementation

4.1.1. Environmental

The benefits of green building for environmental performance will enable people to cope with serious challenges because of uncertainties in climate change scenarios. The building industry has a substantial influence on the environment, which is considered the most important element in mitigating the effect of global warming on humanity [35,36]. The improvements of energy efficiency and environmental performance in buildings are the core of the green building transformation [35]. The green building as a positive performance construction has a minimal impact on the environmental aspect, which can also reduce lifecycle environmental implications [37]. Darko et al. [38] further note that one of the major objectives in green building is to minimize environmental interference and construction waste. The treatment of construction and demolition wastes plays an important role in the development of sustainable building design. The value of recovery rate in construction wastes should be over 90%, which can distinctly reduce the effect of waste generation [39]. Moreover, the significant amount of waste from building can result in air and water pollution [2]. Thus, the choice of building materials plays an important role in sustainable development, and could help to provide healthier and safer surroundings. Timber is an environmentally sustainable material for building construction that leads to reduced energy consumption and $CO_2$ emissions [40].

The life cycle of building construction is associated with energy consumption and greenhouse gas emissions [37,38,41]. The building industry is a "resource-intensive industry" that annually consumes huge quantity of energy and natural resources, including over 40% of energy, 40% of raw materials, 16% of water, and 25% of timber consumed worldwide [37]. It has been further demonstrated that over 40% of worldwide greenhouse gas emissions are from the building industry. The green buildings have great advantages

over traditional buildings, usually bringing higher performance embodied in energy conservation, water conservation, and $CO_2$ emissions reduction [18,38,39]. In fact, the green building can be considered as the specific presentation of energy efficiency and resources efficiency. Architects, along with increasing awareness of climate change, should recognize the energy performance of building designs. Sustainable buildings can conserve 40% more energy compared with traditional buildings, increasing energy efficiency and decreasing $CO_2$ emissions, which signifies that the design should make room for conserving energy and reducing emissions [38]. Additionally, the low-carbon materials in green building have been demonstrated to reduce life-cycle emissions of buildings by up to 30% [42]. Therefore, the green building can be used to realize the sustainable development of low-carbon construction. It is typically designed to achieve the goal of high environmental performance, which can be evaluated through green building tools such as LEED certification [37]. Zuo and Zhao [37] mention that commercial buildings would benefit the most from LEED certification, followed by residential and public buildings. Besides, the energy consumption of buildings in urban areas can be made quantitative by applying a spatial regression model to analyze the connection between building greening rate and surface temperature [36]. Furthermore, green building can also contribute to the enhancement of urban biodiversity and the protection of ecosystems through sustainable land use [43,44].

4.1.2. Economic

The green building can lead to significant economic savings by improving life cycle cost (LCC) methodology and application, particularly from construction, maintenance, and operation sections. LCC includes the entire cost including building design and construction, building maintenance and operation, and building disposal [43,45,46]. Technically, it follows the principle of engineering economics and can be used to complete life cycle budgets involving environmental and social costs during the building's life [37]. There is a definite relationship between cost savings and enhanced building performance from a life cycle viewpoint [46]. Energy consumption can be an important characteristic during building design and construction. Therefore, the focus of energy conservation research has been to assess the degree to which green buildings reduce energy consumption compared with conventional buildings that comply with regulations. Lin et al. [47] investigated the energy consumption of green buildings in China and found that in hot summer and cold winter regions, the average total energy consumption of Type A buildings (mixed mode) and Type B (mechanical conditions) green buildings is close to the recommended values of 60 and 80 kWh/($m^2$a) in the Building Energy Consumption Standard and the energy consumption of green buildings is statistically significantly lower than that of conventional buildings, with the average energy consumption of Type A buildings (mixed mode) and Type B (mechanical conditions) green buildings being 15% and 23% lower than the upper limit required by the Chinese standard, respectively. An average reduction of 16% in carbon footprint can be achieved for commercial buildings through executing energy efficiency methods to boost the life cycle cost performance of the green building [37]. It has been explained that greenhouse gas emissions as a social cost have been monetized by carbon pricing. For consumers and contractors in residential projects, the initial investments of green building could be higher than that of traditional building; however, these investments can be paid monthly by dividends in the form of saving on electricity bills in long run [48].

Cost savings in the maintenance and operation sections might be conducive to counteracting the prior cost of green building construction [19,45]. The project life cycle cost analysis can be used to decide the applicable upfront cost in green building design [43]. Moreover, cost savings by the reduction of energy consumption and less maintenance and operation costs are the major source of revenue for green building management [49]. Zhang et al. also report that the aggregate operation costs consist of water and power utilities, common maintenance, grounds maintenance, garbage recycling, and cleaning costs. Fundamentally, periodic maintenance could extend the building service life to meet the lowest adequate level of performance, which is essential to identify the time range of

the complete building life cycle [43]. Once the time horizon in a building's life cycle is estimated, green building costs such as operating parts can be calculated, which involves direct and indirect benefits because of resource utilization [1]. The improvement of resource utilization can bring direct benefits from energy use intensity and bring indirect benefits from environmental protection, respectively [1,50]. Meanwhile, green buildings should be furnished using advanced integrated strategies and technologies. Optimizing sustainable buildings' operating costs can be done through two approaches: passive design and active design [1,48]. Passive design is the use of airflow and sunlight to provide a relaxing indoor atmosphere while reducing the energy demands of heating, ventilation and air conditioning (HVAC) equipment. Active design is the application of advanced technologies and systems to improve energy efficiency and reduce the resource consumption of building operations. Passive design has evolved along with active design, using advanced technologies to promote the use of sunlight and airflow to reduce the building's energy demand.

### 4.1.3. Social

The social benefit of green building contains corporate social responsibility. Environmentally friendly management might achieve social benefits under the emerging idea of ecological development. One of the main drivers of green building is the demand for corporate social responsibility [1,45]. Numerous companies take voluntary actions to conduct environmental issues such as reducing greenhouse gas emissions as part of their corporate social responsibility, a common situation in Japan [1]. The improved corporate social responsibility can likely promote the creation of a good corporate image in order to develop green construction and sustainable building renovation. Companies in the oil and construction industries that rent or buy a green building might indicate their corporate commitment to the environment and obedience with corporate social responsibility, which can help the enterprises to obtain higher corporate reputations and indirect financial interests. Investors who accept green standards in the project's construction can gain preferential land prices from the local government [37]. It has been further claimed that improving corporate reputation could also make it easier for companies to attract investors. Therefore, the capital costs of companies can be reduced because of the preferable corporate social responsibility performance. Additionally, competitive markets in green building will be formed to effectively stimulate the development of green and sustainable buildings [51].

### 4.1.4. Health and Safety

There are other benefits associated with health and safety if occupants move to green building. Indoor environmental quality (IEQ) is the most critical component of human benefits in green building, which includes lighting and indoor contaminants [37,45,52]. Illumination distribution as one aspect of the indoor lighting environment can affect mental health and work efficiency [1,37]. Indoor air quality plays an important role in the performance of building and participant health [1,37]. Buildings could directly influence human health as residents spend most of their time indoors. The growing effect on cities in determining global environmental and health outcomes has led to a greater emphasis on urban policies to address human health [45]. The green building can realize higher IEQ levels than traditional buildings, improving the health of residents and resulting in increased user satisfaction [1]. Nevertheless, it has been also indicated that the attitude of indoor environmental quality from green building participants is more tolerant than that of traditional participants. Moreover, the sleep quality of occupants in green-certified buildings is 6% higher than that in non-certified buildings [53]. Furthermore, energy savings should not come at the expense of health, which will help close the gap between customer expectations and design solutions for future green building developments. Meanwhile, many governments around the world imposed lockdowns during the COVID-19 pandemic to avoid the spread of the coronavirus. However, lockdown measures also caused widespread mental health problems among urban residents [54]. Through the contribution of the Chinese national Assessment Standard for green building (GB/T 50378-2019) to the

fight against COVID-19, Wang et al. [55] found that green buildings can reduce the risk of infection and prevent cross-infection, promote people's health, and maintain the stability of working life during the epidemic. The contributions of green buildings to the health of the population during COVID-19 are: (1) Green buildings control the concentrations of indoor air pollutants and promote the health of building occupants. (2) Mold can induce diseases such as asthma, allergies, rhinitis, and respiratory infections. Green buildings help to avoid the growth of mold and other pathogenic bacteria caused by dew and condensation on the building envelope, thus ensuring the health of occupants. (3) Green buildings can ensure the safety of water, avoid the health and safety risks caused by the wrong connection of pipes, and reduce the risk of cross-infection caused by the quality of different kinds of water. (4) Green buildings can benefit the health of residents through the use of green building materials with antibacterial functions.

### 4.1.5. The Summary of Benefits of Green Building Implementation

Overall, the implementation of green buildings can effectively reduce carbon dioxide emissions and energy consumption, and the recyclability and low carbon nature of the materials used in green buildings make them a practical solution to environmental problems, as shown in Table 2. At the same time, the life-cycle cost approach allows green buildings to save on electricity costs and thus reduce the cost of building occupancy. The use of green buildings can enhance the social reputation of companies and promote the formation of a competitive market for green buildings. Finally, residents of green buildings have better sleep quality, health and emotional stability.

### 4.2. Challenges to Green Building Implementation
#### 4.2.1. Environmental

The high concentration of populations and economic activity in city regions has enhanced the relations between cities and the environment. Cities play an important role in environmental and health issues, which are also critical for sustainable development in the future [21]. It has been further stated that urban construction environments could bring significant ecological profits with appropriate policies and actions. Therefore, the environmental challenges to green building deserve attention in different elements of urban construction environments. The further improvement of natural ventilation and energy performance are two challenges for green building design, as they are important components reflecting the quality of the design [53,56]. Natural ventilation is an important component to reflect the quality of the design, which becomes an impediment during green building design. The size of the ventilation openings is difficult to decide upon, due to the pressure differences of natural ventilation systems [53]. Natural ventilation can be improved by optimal building design, including air temperatures, relative humidity, air velocity, and mean radiant temperature [57]. However, it is difficult to ensure these parameters in green building design [58]. Thus, the existing studies commonly agree that natural ventilation is one of the impediments to green building design. Actually, energy efficiency in the building sector consists of thermal comfort, energy performance, indoor air quality, HAVC systems, daylighting and creatively integrated energy conservation approaches [59]. The sustainable energy performance of green buildings can be utilized to alleviate greenhouse gas emissions and reduce the energy consumption of the building, which is also associated with significant challenges such as implementation, maintenance and operation costs [1,60,61]. The building sector has become the largest consumer of energy, overtaking the transport and industrial sectors in many regions and countries such as the European Union and the United States [60]. Thus, energy performance can have a direct impact on environmental change in urban areas. Although local governments provide relatively well-developed policies and regulations, and rating systems are popular worldwide, the industry and the public still believe that green buildings are not fulfilling their design promises, including energy efficiency achievements [62,63]. Commercial buildings with LEED certifications do not significantly outperform non-LEED commercial

buildings in energy conservation performance or even display reductions in greenhouse gas emissions related to building operations [12,60]. Some green building projects are designed in accordance with standard regulations but are unreasonable design, which might result in higher energy consumption than non-LEED buildings [64].

**Table 2.** Sustainable benefits of green building implementation.

| Sustainable Indicator | Key Findings/Benefits Related to Green Buildings | References |
|---|---|---|
| Environmental benefits | ■ High overall energy performance, water conservation and reduction of $CO_2$ emissions. Sustainable green buildings can reduce energy consumption by up to 40% to reduce $CO_2$ emissions.<br>■ The waste recycling rate in green buildings can reach more than 90%, which can significantly reduce environmental pollution.<br>■ Use of low-carbon materials (such as timber) has been shown to minimize life cycle carbon emissions<br>■ Contribute to urban biodiversity enhancement and ecosystem conservation through optimal land use strategies in sustainable development. | [18,36–39] |
| Economic benefits | ■ Significant economic savings can be achieved through improved life cycle cost (LCC) methods<br>■ Investments on green buildings can be effective through long-term electricity cost savings. | [43–45,51] |
| Social benefits | ■ Construction companies can enhance corporate reputation and attract potential investors easily<br>■ Improved asset value due to perceived high user satisfaction and social performance benefits | [1,37,51] |
| Health and safety benefits | ■ Achieve higher levels of indoor environmental quality (IEQ), improve health of residents and increase user satisfaction<br>■ Residents are more emotionally stable than those who live in traditional buildings<br>■ Residents have 6% higher sleep quality than those in non-certified buildings.<br>■ Green building applications can control indoor air pollutants during the COVID-19 pandemic, inhibit the growth of pathogenic bacteria and ensure water safety to avoid cross-contamination. | [1,53,55] |

### 4.2.2. Economic

The green building needs to address not only the environmental challenges but also the economic impediments to the process of sustainable development. Economy occupies the central position in the sustainability of urban development in many countries [65]. The green building is regarded as the fundamental component in sustainable construction development [66,67]. The construction industry plays an undeniable role in economic growth because of its high contribution to the global Gross Domestic Product (GDP), about 8–10% [1]. While green building provides a wide range of benefits to society, it also faces various economic challenges during the development process. The most important barriers to green building development can be divided into two aspects: higher initial investment costs and a lack of market demand [7,46,68,69]. Higher initial costs are perceived as one obstacle to the widespread adaptation of green buildings in the construction industry [70–74]. Innovation technologies often come with higher installation costs, which is one of the main reasons for the lack of energy efficient buildings [70,75]. However, what is usually overlooked by decision makers and stakeholders is that the building has the characteristics

of durable structure and long-term investment. They often consider upfront costs and initial costs when determining building design schemes [76]. High initial construction costs have an impact on the potential of the green building to achieve sustainable development. The construction costs of green building are approximately USD $10/m^2$ to USD $30/m^2$ compared with that of conventional building [75]. Generally, green buildings with higher LEED certification levels involve more construction costs. A case study in Thailand indicates that the construction costs of green building at different certified levels, such as silver, gold and platinum, are 0.23%, 1.21% and 6.62% higher than traditional building, respectively [77]. Nevertheless, it is also found that, especially in the tropics, the operating cost of a building can be reduced due to investment in energy-saving technologies and other green functions. Energy consumption in green building design is approximately 34% of that of traditionally designed design [75]. As the energy consumption in the operating stage is relatively large in the tropics, the rate of energy consumption reduction is generally high [78]. Additionally, over 75% of green buildings have initial extra construction costs that are between 0% and 4% [79]. It has been further stated that whether the cost saving from building operation can cover the initial extra construction cost is still under investigation. The extra construction cost might result in cost overruns and a higher final price during the life cycle cost process [7,12,60,80,81]. A study reveals that the life cycle cost of green building projects in Indonesia is raised by 13.75% to obtain platinum certification [82]. Furthermore, major impediments in developing green building construction are related not only to higher initial investment costs, but also to a lack of market demand and lower cost-effectiveness.

The other major obstacle to the adoption of green building is a lack of market demand [12,83]. The construction industry is an important driver of economic growth. Green building technology might be applied in the planning, procurement, design, construction and operation stages, which shape the construction industry in many countries [83]. As different countries have different market demands for green buildings, this aspect can provide both opportunities and threats in developing the green building market. Moreover, the development of this market could have a positive influence on the succedent application of green building. It is important to explore the market situation and its related variables, such as low capitalization and poor organizational structures, which play a vital role in the development of the green building market [84]. According to Addy et al. [12], one of the most important socio-economic barriers for many developing countries is the supply of housing. Housing shortages and poor infrastructure could retard the development of the building industry. Moreover, the price of residential buildings built by the private company is not controlled by the government, but by its demand and supply [3]. The additional cost of sustainable buildings, such as LEED certification, design cost and the resulting low profit margin, are the main disadvantages that hinder the practical application of sustainable buildings [12,85]. It has been further reported that the economic crisis and general economic downturn in the past decade have exerted constant pressure on construction projects, resulting in a moderate increase in construction costs. The major concern of the building industry is the financial and operation capital of contractors and investors [86]. They will only pursue green building projects if there are sufficient profit records. Thus, most investors have less desire to consider capital-intensive construction projects with increased uncertainty in investment costs [3,70,87]. It has been suggested that the structure and performance of the market determine how easy it is for capital to enter or leave the market, which eventually contributes to the low demand for the green building market. Therefore, identifying these issues might help to enhance the competitiveness of the green building market with increasing building industry competition across continents while ensuring more sustainable economic development and less environmental disturbance. The potential market stakeholders and government can absorb and utilize the information of these green buildings at both the socio-cultural and institutional levels to improve the reach of sustainable development in urban areas.

### 4.2.3. Socio-Cultural

The consumers and public are the main participants in green building development. Lack of awareness about green building and lack of public participation are two main obstacles in the socio-cultural aspect [12,68,72,80,88]. Some green and sustainable strategies are emerging to mitigate some of the negative impacts on the environment. The construction industry is moving towards greater energy efficiency. Nevertheless, one of the biggest impediments is a lack of awareness about green building among contractors, homeowners, and clients [12,67,79,87,88]. A lack of overall awareness and understanding of sustainable development issues hinders the realization of sustainable construction goals in real time [72]. Although the theory of green building has been demonstrated for decades, the basic comprehension of green building among customers is still weak [12]. Awareness among construction professionals has increased, but the information available to the customer is limited and sometimes misleading [89]. The customers lack related information to select more sustainable projects in most cases [90]. If there are regulatory gaps about the inappropriate information of green building, real estate developers and other market participants might adopt opportunistic behavior and avoid providing real green building projects [12]. There is no adequate understanding and awareness of sustainable development issues for contractors, clients, and other public organizations [80]. Moreover, the concept of sustainability is still relatively new for many developing countries, which causes the exploitation and utilization of sustainable buildings such as green buildings to fall short of the maximum social benefits [80,87]. Additionally, some stakeholders are reluctant to change due to an inadequate understanding of the concept of green building construction [68,88]. Therefore, social and cognitive barriers to green building constitute a major barrier in the construction industry. Higher social performance through sustainability is difficult to reach in urban areas.

A lack of public participation is another challenge in green building construction [68,80]. Traditional urban development planning is a top-down process during decision making. Although sustainable urban planning is a bottom-up process, the effect of actual implementation is not obvious to all participants and stakeholders throughout the whole planning process [88,91,92]. The decrease in public participation might indirectly affect public attitudes and behaviors towards green building products [88,93]. In this era of construction, human health and comfort should be taken into account during the development of green building, which can be reflected in public attitudes and behaviors [83,94]. This is because the public will establish basic evaluation criteria based on the collected information to guide the comparison and assessment of some potential green building products. Addy et al. [12] further point out that green building contributes an integrated industrial chain with the mutual support of different sectors from production to operation. However, from the perspective of buying and selling, consumers and real estate investors are the major participants in the rapid and healthy development of green building [67]. Therefore, increasing public participation is necessary after analyzing the attitudes and demand requirements of consumers in green building. Additionally, it is essential to study the social acceptance of green buildings to adequately consider public interests and opinions in order to reflect the preferences and potential influence behaviors in green building.

### 4.2.4. Institutional

The most important impediments to green building development in its institutional aspect can be summarized in three parts: a lack of government financial incentives, deficient education and training regulations, and a fragmented legal and institutional framework [1,67,79,95–97]. The role of governments is undeniable and effective in promoting green building development according to many experts [60,98]. Institutions and regulations should be enforced to develop the green building industry. Furthermore, the government can facilitate the popularization of green building through various approaches. However, there is an institutional challenge that is a lack of government financial incentives, which likely impedes the development of green building [3,38,60,67,68,80]. Generally, tax ex-

emptions can be introduced as a government incentive to encourage capital investment in green building. Nevertheless, it is found that the current incentive measures are not enough to effectively encourage building construction enterprises to devote themselves to the development of green building [98]. Moreover, Samari et al. [67] further point out that construction companies cannot recover their high upfront costs of green building planning with government financial incentives. Therefore, the necessary institutions and regulations should be developed to support government financial incentives, which might provide affordable financial resources to lower the investment risk for commercial and residential construction developers. Additionally, as most of the policy and economic support is aimed at developers, architects and designers are likely less motivated [12,79]. Architects are the initial designers and participants in green building construction, which plays an important role in deciding the basic features and performance of the building. The personal attitudes of architects such as social responsibility and enthusiasm for green building applications are critical to the implementation of green building. Thus, architects can also act as promoters of green building to enhance its social recognition.

Lack of education and training regulations is also a key barrier to the development of green building [60,67]. Education level could influence the awareness of professionals about green building. According to the research of Samari et al. [67], awareness about green building will be increased with the improvement of the education level of participants. Therefore, improving the education level of the public will increase interest in the green building market by construction companies. More environmentally friendly approaches might be adopted in future projects, which can be regarded as part of corporate social responsibility [79]. Because of the high attention paid to the environmental impact of projects, the utilization of green facilities in buildings by professionals will increase significantly. Furthermore, the lack of training programs is not conducive to the promotion of green building knowledge [1,99]. This obstacle could possibly generate a lack of preparedness of the government in implementation because of incomprehensible technical guidelines and regulations [71].

The fragmented legal and institutional framework is another major obstacle [1,79,95]. If different government departments develop legal and institutional frameworks, it will make the whole system complicated. The division of responsibilities among various departments is not effective [12]. Therefore, institutional complexity is seen as hindering the efficiency objectives of the policy and legal frameworks established by local governments. Additionally, it is believed that the implementation of a regulatory framework and mandatory requirements for sustainable design will promote the development of sustainable performance most effectively [100].

### 4.2.5. Policy

The effectiveness of the current polices barriers is crucial to the development of green building [68,100]. The public has gradually questioned the effectiveness of some policies in the implementation of the current policies [101]. The current green building evaluation policies are not equitable for different cities with regional differences in economic development. The investigation found that in advanced cities in China such as Beijing and Shanghai, the evaluation criteria are considered to be extremely low, and many projects can easily satisfy the requirements. On the contrary, for regions in China that are lagging behind in economic development, such as Guizhou and Gansu, the standards of these requirements are so high that few projects can achieve the requirements [102]. Access to environmental information on large-scale construction projects for the public is still limited; developers usually consult experts to satisfy the legal requirements [12,103]. Enough attention should be paid to negative feedback from the public during the process of policy revision and formulation in the future. Additionally, legislation and tougher penalties for non-compliance are appropriate ways to promote green building development [104].

### 4.2.6. Technological

A lack of technology, such as Building Information Modeling (BIM) is also a major barrier in the process of green building development [103,105–107]. BIM is a popular application for architecture, structure and construction [107]. It has been further stated that the application of BIM technology is conducive to improving the efficiency of the building industry and assessing the sustainability of green buildings. However, high upfront costs and poor inter-software interoperability impede the development of BIM technology, which causes the low popularity of this tool in some projects [103,107]. Different companies apply different software, which makes it difficult for them to design the same project using various operating platforms [108]. Different software can likely lead to information transmission deviation. Product information would be lost, resulting in communication problems among different design departments. Therefore, it is necessary to update the design software in green building, which can promote the development of the construction industry. Moreover, their associated rating systems can also influence the green building design [2,104]. The main factors driving the growth in demand for green rating systems are the global pressure for sustainable development, social and end-consumer pressures, and government pressure on compliance [104]. Furthermore, bottlenecks of green building have not been overcome due to a lack of key technologies such as energy-saving research [2]. New technologies influence the speed of transformation of the construction market towards sustainable aspects. The complete technology system in green building construction should be improved, which could create an effective platform to promote and communicate new technologies.

### 4.2.7. Summary of Challenges of Green Building Implementation

Overall, there are many contemporary challenges to the implementation of green buildings, as shown in Table 3. The optimal design of green buildings needs to be considered in all aspects, otherwise any deviations in design can lead to higher total energy consumption and result in a burden on the environment. The main economic challenges of promoting green buildings include the high initial costs and high capital investments and the high investment required for research and development to benchmark innovative practices. In addition, the use of green buildings requires collective efforts from all participants and stakeholders to thoroughly understand the technical guidelines and regulations of green buildings and to promote green building implementation. Furthermore, green building evaluation policies need to be further improved according to regional economic development differences. In the construction of green buildings, the iteration of new and emerging technologies can lead to information bias among all parties involved in the construction process and research on key technologies for energy efficiency needs to be further developed.

**Table 3.** Challenges of green building implementation.

| Aspects | Key Findings | | References |
|---|---|---|---|
| Environmental challenges | ■ | Air temperature, relative humidity, air velocity and average radiation temperature are difficult to control in green building design. | [21,64] |
| | ■ | Green buildings designed according to the standard provisions with poor design issues may lead to higher overall energy consumption than non-LEED buildings. | |

**Table 3.** *Cont.*

| Aspects | Key Findings | References |
|---|---|---|
| Economic challenges | ■ Higher initial costs are as a barrier to the widespread adaptation of green buildings in the construction industry. The construction cost of green buildings can be as high as about USD $10/m^2$ to USD $30/m^2$. <br> ■ The construction costs of green buildings with different LEED certification levels, such as silver, gold, and platinum, are 0.23%, 1.21%, and 6.62% higher than those of conventional buildings. <br> ■ Lack of market demand in different countries. <br> ■ Additional costs of sustainable green buildings, such as LEED certification and design costs contributing to low construction profitability | [12,70,71,74,75,77,83,85] |
| Social-cultural challenges | ■ Lack of awareness and overall sustainability knowledge and minimum understanding of long-term benefits among contractors and owners <br> ■ Collaboration among all participants and stakeholders in the green building scale-up process is ineffective and lacks public participation. | [12,67,72,79,87,88,91] |
| Institutional challenges | ■ Lack of government financial incentives has hindered the promotion of green buildings. Current incentives are not sufficient to effectively motivate building construction companies to actively invest in green buildings. <br> ■ Lack of education and training programs. <br> ■ The fragmentation of legal and institutional frameworks presents a challenge for green building development. Proper articulation is necessary to mitigate this challenge. | [1,3,12,38,60,67,71,79,105] |
| Policy challenges | ■ Current green building evaluation policies do not address the priorities in different cities and considerations of regional differences with economic developments. <br> ■ Minimum legislations and penalties for non-compliance practices in the green building sector. | [100,102,104] |
| Technological challenges | ■ The integration of new tools such as Building Information Modeling (BIM) technology in green building projects is not widespread. <br> ■ Application of different software by different companies can result in deviations in information transmission, resulting in communication problems between different design departments. <br> ■ Lack of key technologies, such as energy efficiency research in green buildings. | [2,103–107] |

## 5. Current Advancements in Green Buildings

The development of green building presents an opportunity to accomplish low-carbon construction and energy efficiency. The potential of green building development is great, and it is considered as an important industry across the world. The improvements of environmental performance and energy efficiency in buildings are the core of green building transformation [36,109]. The building industry has a substantial influence on the environment, which is considered the most important element in mitigating the effect of

global warming on humanity [6,39,41,110]. Chi et al. [111] put forward that construction waste minimization performance of green building projects in the US and China presents a significant difference at lower certification levels. The reason might be a low reuse rate of construction components in China so that the construction waste minimization performance of green building projects is influenced. However, the study would have been more interesting if it has included the construction waste minimization performance of green building projects in economies other than the US and China under different political, economic, social, and technological conditions. In fact, green building can be considered as the specific presentation of energy efficiency and resources efficiency. Some findings from studies support the idea that green building has great advantages over traditional buildings, which usually bring higher performance embodied in energy conservation, water conservation, and $CO_2$ emissions reduction [28,37,38,112]. Additionally, the energy consumption of green buildings in urban areas can be made quantitative by applying a spatial regression model to analyze the connection between building greening rate and surface temperature [42]. Therefore, green building can be used to realize the sustainable development of low-carbon construction and energy efficiency.

Building regulations have not been well developed; this will lead to performance disparity in green building. Conversely, the development of green building will promote the improvement of building regulations. Building regulations must be constantly updated to keep pace with the development of green building construction technology. For instance, China's green building standards and regulations are constantly improving. Local standards are amended every few years under the guidance of National standards by combining the lessons learned during the operation and working experience. In 2019, China released the latest Assessment Standard for green building, 2019 version. Compared with the old version (2014 version), this one is updated for the latest technology and concepts of green building, and further clarifies the misunderstandings in the 2014 version. The introduction of the 2019 version of green building standards reduces the exploitation of loopholes and regulates the behavior of relevant stakeholders. Nowadays, many institutions propose supportive green financial policies to promote the development of green building. These include Basic Concept and Crucial Advocacy of Sustainable Finance (ISO/TR32220:2021) released by International Organization for Standardization in 2021, Action Plan: Financing Sustainable Development proposed by European Union in 2018, Equator Principles from several united international banks, Green Bond Principle from International Capital Market Association, and Climate Bond Standard developed by Climate Bond Initiative. The government of the Philippines has passed the Green Jobs Act in 2016, recognizing the potential of the green building industry to deliver sustainable economic development and reducing related tax for green building companies [28,113].

The green building design often relies on budget limitations and health and safety factors and cannot be limited to reducing environmental impacts alone. As almost all construction companies seek profits, achieving economic benefits through construction is an essential factor that the contractors look into when making a decision to construct a green building. Fundamentally, periodic maintenance could extend the building service life to meet the lowest adequate level of performance, which is essential to identify the time range of the complete building life cycle [43]. Once the time horizon in a building's life cycle is estimated, green building costs such as operating parts can be calculated, which involves direct and indirect benefits because of resource utilization [49]. The improvement of resource utilization can bring direct benefits from energy use intensity and bring indirect benefits from environmental protection, respectively [1,43,50]. Meanwhile, green buildings should be furnished with advanced integrated strategies and technologies. More specifically, the sustainability objectives of green building construction and operation are to minimize environmental influence and improve resource utilization as well as to maximize the effectiveness and return on investment of each building component.

## 6. Future Recommendations

Based on the previously proposed impediments to green building and the current development of green building, it is proposed that some directions and specific measures for the future improvement of green building are proposed. The technology system of green building construction should be accelerated to adapt to market demand in the future. Developers should strengthen the research and development of key technologies [96]. The government should increase support for technological research and communication in order to improve the commercialization level and market penetration capacity of new technologies. Meanwhile, innovative technologies need to be commercialized through an effective platform to demonstrate and promote them.

Property developers are the decisive stakeholders in green building development activity to decide whether to exploit green technology, products, and projects. Improving the awareness of the stakeholders is helpful for the process of green building development. The purpose of most developers is to pursue profit. Even developing green building is a strategy to gain the benefit, and not only the profit, but also enterprises' future reputation and social status. However, to obtain future honor without compromising the current payment collection is a precondition, especially for small-scale building companies. To achieve sustainable building requires high-quality building elements for improving building performance. For example, in winter, heat energy can be saved by increasing the windows' air impermeability; good sound insulation is necessary to reduce interior noise. Thus, windows with better performance are needed, which will increase the initial cost. Under this circumstance, consumers seriously consider good cost performance instead of sustainability [2]. A fact is that small-scale companies are only capable of adopting basic measures for building development, while only large-scale companies have enough money to consider green building development. Therefore, it is necessary to take measures to enhance the sustainability awareness of stakeholders from small-scale companies; it is their responsibility to propagandize and popularize green building, and further establish a mature green building market. In return, the market will attract and support more companies to develop green building.

The government can provide campaigns at the local, state and national level to promote the environmental, economic, and social benefits of green buildings. Moreover, governments could also conduct awareness programs to encourage developers and tenants to accept green building because of the increasing value of such buildings, lower operation costs, and better community image. The incentive policies in finance, taxation and economy should be increased. These incentive policies might generate a strong internal motivation for relevant departments to stimulate the development of green building. Green finance is a special financial activity that is used to support sustainable development. Projects that aimed for developing green building can apply for financial support and lessen the financial burden. Therefore, the government could repay some benefits such as tax cuts to the developers and consumers if the invested project is green building. Related regulations and policies should be improved in the initial stage for some countries and regions. Legislation is the basis of promoting green building [12]. Although many standards about green building have been issued for many years, the supposed effect has not been achieved, such as energy efficiency standards [2]. Penalties should be imposed for projects that do not meet mandatory standards. The standards of green building should satisfy the local conditions to provide direct guidance. Furthermore, the establishment and improvement of green building standards should adequately consider local economic development, weather conditions, regional resources, and construction level. Additionally, the evaluation system should contain high quantitative criteria. The research and application of quantitative and qualitative indicators should be strengthened. Only in this way can green building be really 'GREEN'. Furthermore, the lifecycle evaluation of green building should be completed before implementing the design proposal and construction process.

## 7. Conclusions

The current study provides a detailed review of the consideration of green building to achieve SDGs. A large number of definitions of green building are expressed according to different characteristics such as general definition, energy and resource focused definition, and comprehensive definition. The green building achieves good performance in energy conservation and reducing $CO_2$ emissions. The development of green building will promote the improvement of building regulations. The evaluation criteria can support the development of green building and be conducive to the goals of cost efficiency and project sustainability. The social advancement of green building contains the formation of corporate social responsibility.

Meanwhile, green building can face economic implications during the development process, such as high initial investment costs and a lack of market demand. Additionally, a lack of awareness and public participation in green building are other social obstacles. The main institutional obstacles to green building development can be summarized as the lack of government financial incentives and fragmentation of legal and institutional framework. A lack of potential to enable effective collaboration with the various stakeholders is also a major obstacle to the development of green buildings. Therefore, the technological system of green building construction should be accelerated to adapt to the market demand in future. It is crucial to improve the awareness of stakeholders to promote the development process of green building. The government should launch campaigns to encourage developers and tenants to embrace green building, which can add value to buildings. The incentive policies might become a strong internal motivation to boost the popularity of green building. Additionally, legislation is a fundamental element to promote green building. The local economic development, weather conditions, regional resources, and construction level should be fully considered during the establishment and improvement of green building standards. Future studies should be focused on comparisons of green building performances among different counties under several categorizations. Although the implementation of green building faces various challenges, green building has a great potential to reduce energy consumption and the greenhouse effect.

**Funding:** This research received no external funding.

**Institutional Review Board Statement:** Not applicable.

**Informed Consent Statement:** Not applicable.

**Conflicts of Interest:** The authors declare no conflict of interest.

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
