# Peer review of "Sustainability Considerations of Green Buildings: A Detailed Overview on Current Advancements and Future Considerations"

_sustainability, doi:10.3390/su142114393_

Round 1
Reviewer 1 Report
This paper provides a systematic review on sustainable considerations of green building. This study was meaningful, but needs some improvement:
1. The Abstract should contain answers to the following questions: What problem was studied and why is it important? What methods were used? What are the important results? What conclusions can be drawn from the results? What is the novelty of the work and where does it go beyond previous efforts in the literature? Please include specific results in the Abstract.
2. The originality of the paper needs to be stated clearly.
3. Figure 2 shows the research hotspots for green buildings by VOS viewer. It is better to give a detailed explanation for the research hotspots. For example, which hotspots belong to environmental sustainability.
4. The conclusion part is huge and should be summarized with the main important point of results conducted from this study.
Author Response
Authors’ would like to acknowledge all the reviewer’s for providing valuable comments to improve the quality of the manuscript. The following table provides a detailed description of the responses to reviewer’s suggestions and the revisions applied in the revised manuscript.

Reviewer 2 Report
Thank you for the opportunity to review this interesting paper. My comments are below
Green Building is used as the main term throughout the work, but Sustainable Building is, to me, more inclusive of social, economic and environmental issues - I appreciate different regions of the world use differing terminology
Line 52 talks about reducing environmental burdens - this would align with my understanding of the focus of green building, but elsewhere social, economic and environmental matters are discussed - to me this is the wider 'sustainable built environment' that is being discussed
I can see that there is some good discussion about this at the start of section 2, this satisfies my query, but perhaps the issue could be noted in section one, pointing the reader to this discussion in section 2.
Line 116 should state full title of WoS before using the shortened version (swap with line 125)
Figure 2 is helpful - but could explain in the preceding text how endnote was used
For Figure 2, what do the colours and sizes mean? I can guess that larger is more prevalent, but I dont know what the colour means - if would be good to explain
Conducting a literature based paper is challenging, I like the methodological approach here, it is clear and justified
There is a good deal of unbroken text in lines 330-600, breaking the text up a little with a couple of relevant figures or other visual representations may help the reader, likewise 608-950
Chapter 6 would benefit from some tabulation of the barriers discussed - again to help break up the text, but also to give the reader some visual aids
Chapter 7 might also benefit from visual aid perhaps to explain both benefits and barriers along with what should be done about it.
Overall I like the methodological approach, but feel that the 'so what' aspects of the work, particularly in sections 6 and 7 could be improved - what do we have now that we didn't before this work came out?
The idea of a literature base paper should be to draw out key themes, arguments and challenges, I think any revision to the work could focus on adding emphasis to these aspects.
Author Response

(The authors gave the same response as above.)

Reviewer 3 Report
Dear authors. I'm willing to admit that you have done a fairly thorough review of the literature related to so-called "green" building. However, your work does not even contain the primary signs of your own scientific research. It is no coincidence that the conclusions are a set of trivial truths and statements on the topic of environmental protection and energy conservation. I don't consider Sustainability a place for review posts. If you consider the methodology of bibliographic analysis to be such an element, then Sustainability is not a publication corresponding to the profile of such studies. Thus, the decision remains with the editors.
Author Response

(The authors gave the same response as above.)

Reviewer 4 Report
The paper presents a literature review on the environmental, economic and societal impacts of green buildings and the barriers to their development. The study focuses on the analysis of a corpus of papers from the web of science database published between 2015 and 2022.
Even this state of the art is based on papers search between 2015 and 2022, older references (2007, 2010...) are found in the document.
The key word used are « five key words were used; green building, sustainable building, environmental impacts, economic benefits, and social impacts were linked with “OR” operator to capture the maximum result”
I think, instead to use “OR” to link, you should use “and” to link Green building and social impact or economic impact...
What about impact on comfort, well-being and healt of occupants?
The authors claimed “The results of the literature measurement assessment indicate that the main buzzwords in the field of green building include environmental impacts of green building, cost-effectiveness of green building, capabilities of green building, manufacturing of green building as well as barriers and challenges to green building.”
Do these key words provided by the VOSviewers assessement? How the authors combined them with the first list of key words?
In figure 2, there's no direct relation between sustainable building and environment, neither environmental impact, how the authors explain that?
Page 9: the authors claimed “Therefore, the focus of energy conservation research has been to assess the degree to which green buildings reduce energy consumption compared with conventional buildings that comply with regulations. Green building can consume 30% energy less than the conventional building (Kim et al., 2014).” as there’s no standard definition of green building, it is difficult to talk about comparison with conventional buildings that comply with the regulations. You need to define what you mean by a conventional building. In France for example, the RT2012 regulation required all new buildings to consume less than 50kWh/year and the new environmental regulation RE2020 has the same requirement and takes into account life cycle analysis. Are we talking in this case about conventional buildings or green buildings?
Page10: The direct link between two strategies, active and passive, and cost reduction is not established in the paper. The active strategy also aims to have relaxed indoor environments
Section 4.1.3 does not deal at all with the social impact of this type of buildings on the inhabitants!
Section 4.5.1: I would be curious to know how emotional stability was measured between conventional and green buildings.
In the analysis of health benefits, it would have been good to integrate work on COVID 19 for example https://covid.uia-architectes.org/wp-content/uploads/2020/05/China_Green_Building_Covid-19_paper.pdf
In the environmental challenges, the authors mention natural ventilation as a major challenge. Since they have shown before that these buildings consume less energy and are more comfortable, why should NV be a challenge and a problem?
It would have been nice in the different analyses to distinguish between winter and summer since the behavior (impact, benefit and challenge) could be different according to the season
There is a lot of redundancy, e.g. section 6.1 that has already been dealt with in paragraph 2.4.1.
It would have been nice to compare the progress in the field on green buildings against an older paper review (for example: A bibliometric review of green building research 2000-2016, X Zhao, J Zuo, G Wu, C Huang - Architectural Science Review, 2019 - Taylor & Francis).
Generally speaking, I found the paper to be well structured but with many redundancies. However, it lacks hindsight and analysis, especially on the recommendations.
Author Response

(The authors gave the same response as above.)

Reviewer 5 Report
The structure of the paper appears to be quite complex. The paragraphs "4.1 Benefits of green building implementation", "4.2. Challenges of green building implementation" but partly also the paragraphs "5. Current advancements in green buildings" and "6. Impediments and future recommendations" express the same concepts many times , although in some cases from different points of view. It is advisable to synthesize, eliminating repetitions.
The sentence on lines 398, 399, 400 and 401 corresponds to the sentence on lines 770, 771, 772 and 773.
The paper does not clarify what the investigated context is. USA and China? It would be advisable to clearly state right from the introduction to which geographical context the survey carried out refers. It follows that when the paper indicates local governments as actors in the promotion of green buildings, the reader is not oriented. We can apply the same considerations to the benefits and challenges of "green building implementation".
Author Response

(The authors gave the same response as above.)

Round 2
Reviewer 3 Report
Dear authors, you have done a good job of taking into account my comments.
Author Response
Thank you for the acknowledgement of the revisions completed by the team
Reviewer 4 Report
Please add correctly the reference page 22 line 613.
Author Response
The authors would like to thank the reviewer for pointing the reference issue. We have changed the Table 3 cross-reference to the correct form. And once again, we have checked and corrected the cross-references in the full paper.
Reviewer 5 Report
The structure of the paper continues to appear to be quite complex and redundant.
Although the authors have done a good review of the paragraphs "4.1 Benefits of green building implementation", "4.2. Challenges of green building implementation", "5. Current advancements in green buildings" and "6. Impediments and future recommendations", in my opinion, there is still a certain overlapping of themes and issues.
I suggest the authors to review the contents of paragraph 6 in such a way as not to repeat the same arguments already present in the paragraphs 4.1, 4.2 and 5. In this way the paper would have a clearer articulation. In fact, part of the contents of "Impediments and future recommendations" could be better moved in "benefits", in "challanges" and in "current advancements" too.
Author Response
Authors’ would like to acknowledge all the reviewer’s for the 2nd round of comments to improve the quality of the manuscript. attached provide a deteild expalantion for the comments provided and revisions undertaken.
